# Hyperhidrosis, Endoscopic Thoracic Sympathectomy, and Cardiovascular Outcomes: A Cohort Study Based on the Korean Health Insurance Review and Assessment Service Database

**DOI:** 10.3390/ijerph16203925

**Published:** 2019-10-15

**Authors:** Jae-Min Park, Duk Hwan Moon, Hye Sun Lee, Ju-young Park, Ji-Won Lee, Sungsoo Lee

**Affiliations:** 1Department of Health Promotion, Severance Hospital, 10 Tongil-ro, Jung-gu, Seoul 04527, Korea; milkcandy@yuhs.ac; 2Department of Medicine, Graduate School of Medicine, Yonsei University, 50-1 Yonsei-ro, Seodaemun-gu, Seoul 03722, Korea; 3Department of Thoracic and Cardiovascular Surgery, Gangnam Severance Hospital, Yonsei University College of Medicine, 211 Eonju-ro, Gangnam-gu, Seoul 06273, Korea; pupupuck@yuhs.ac; 4Biostatistics Collaboration Unit, Yonsei University College of Medicine, 211 Eonju-ro, Gangnam-gu, Seoul 06273, Korea; HSLEE1@yuhs.ac (H.S.L.); jystat@yuhs.ac (J.-y.P.); 5Department of Family Medicine, Gangnam Severance Hospital, Yonsei University College of Medicine, 211 Eonju-ro, Gangnam-gu, Seoul 06273, Korea

**Keywords:** cardiovascular diseases, endoscopic thoracic sympathectomy, hyperhidrosis, sympathetic overactivity

## Abstract

Sympathetic overactivity is associated with hyperhidrosis and cardiovascular diseases. Endoscopic thoracic sympathectomy (ETS) is a treatment for hyperhidrosis. We aimed to compare the risk for cardiovascular events between individuals with and without hyperhidrosis and investigate the effects of ETS on cardiovascular outcomes. We conducted a nationwide population-based cohort study using data acquired from the Korean Health Insurance Review and Assessment Service. Subjects newly diagnosed with hyperhidrosis in 2010 were identified and divided into two groups according to whether or not they underwent ETS. Propensity scores were calculated using a logistic regression model to match hyperhidrosis patients with control subjects. Combined cardiovascular events were defined as stroke and ischemic heart diseases. Subjects were followed up until the first cardiovascular event or 31 December 2017. The risk for cardiovascular events with hyperhidrosis and ETS was analyzed using Cox proportional hazards regression analysis. The risk for stroke was significantly higher in the hyperhidrosis group than in the control group (hazard ratio (HR), 1.28; 95% confidence interval (CI), 1.08–1.51); nonetheless, no significant difference in the risk for ischemic heart diseases was observed between the hyperhidrosis group and the control group (HR, 1.17; 95% CI, 0.99–1.31). Hyperhidrosis patients who did not undergo ETS were at significantly higher risk for cardiovascular events than the control group (HR, 1.28; 95% CI, 1.13–1.45). However, no significant difference in the risk for cardiovascular events was observed between hyperhidrosis patients who underwent ETS and the control group. Hyperhidrosis increases the risk for cardiovascular events. ETS could reduce this risk and needs to be considered for high-risk patients with cardiovascular diseases.

## 1. Introduction

Hyperhidrosis is a pathologic condition characterized by excessive sweating beyond physiological needs. Hyperhidrosis can lead to substantial impairment in the quality of life [1] and has an estimated prevalence rate of approximately 2.8% (7.8 million individuals) in the United States [2]. The pathogenesis of this condition remains incompletely understood. Nonetheless, primary hyperhidrosis, defined as inordinate sweating with an unknown identifiable cause, has been suggested to be due to sympathetic nervous system overactivity [3,4,5].

Accumulating evidence has indicated the association between sympathetic nervous system overactivity and cardiovascular disease development and progression [6,7,8,9]. Sympathetic overactivity plays a key role in the induction and maintenance of hypertension [6,10,11]. Among congestive heart failure patients, individuals with sympathetic overactivity have been reported to have higher mortality rates than those without it [12,13].

Endoscopic thoracic sympathectomy (ETS) is a surgical procedure involving the transection and clamping of the upper thoracic chain of the sympathetic nerve trunk and is considered one of the treatment methods for primary hyperhidrosis to reduce sympathetic overactivity [14,15]. After ETS, hyperhidrosis patients have been observed to exhibit a reduction in noradrenaline and adrenaline levels, heart rate, blood pressure, cardiac index, and myocardial oxygen demand [16]. A recent epidemiological study has reported that hyperhidrosis patients who underwent ETS had a reduced risk of cardiovascular events [17]. However, few studies have investigated the relationship between cardiovascular disease and hyperhidrosis and evaluated the effect of ETS on cardiovascular outcomes in hyperhidrosis patients. The present study aimed to compare the incidence and risk of cardiovascular events between individuals with and without hyperhidrosis and to investigate the effects of ETS on cardiovascular outcomes in the Korean population. We conducted a large retrospective cohort study using a claims database that includes the entire population of South Korea.

## 2. Materials and Methods

### 2.1. Data Sources

Data were acquired from the Korean Health Insurance Review and Assessment (HIRA) Service. Approximately 97% of the Korean population is covered by the Korean National Health Insurance, whereas the remaining approximately 3% who cannot afford national insurance are covered by the Medical Aid Program [18]. The HIRA, a central office in the Korean Ministry of Health and Welfare, was established to review claims data and assess the medical care quality in South Korea. Furthermore, the HIRA is responsible for maintaining an electronic database that provides information on inpatient and outpatient visits in all Korean healthcare institutions, healthcare billing, and reimbursement claims submitted to the Korean National Health Insurance and Medical Aid Program [19]. The HIRA database contains enormous epidemiological information on demographics, diagnoses, and medical services (procedures and operations), and the diagnosis codes are standardized in accordance with the 7th version of the Korean Classification of Disease, which is a modified version of the 10th edition of the International Classification of Diseases. 

The study protocol conformed to the ethical guidelines of the 1975 Declaration of Helsinki, as reflected in a priori approval by the institutional review board of Yonsei University Gangnam Severance Hospital (institutional review board number 3-2018-0195). The requirement for the acquisition of informed consent from patients was waived because the present study was a retrospective observational study with an anonymized dataset.

### 2.2. Study Population and Design

In this retrospective cohort study, de-identified data on health claims from 1 January 2007 to 31 December 2017 were extracted from the HIRA database. We reviewed the data recorded in the HIRA database and identified patients with primary hyperhidrosis (diagnosis code: R61) who underwent ETS (procedure codes: S4832 and LA361). Combined cardiovascular events were defined as stroke (I60–I64) and ischemic heart diseases (I21–I25). Stroke included hemorrhagic stroke (I60–I62), ischemic stroke (I63), and unspecified stroke (I64), whereas ischemic heart diseases included acute myocardial infarction (I21–I23) and ischemic heart disease (I24–I25).

Following the exclusion of hyperhidrosis patients with cardiovascular events (stroke and ischemic heart diseases) prior to 2010, individuals were divided into two groups according to whether or not they were diagnosed with hyperhidrosis in 2010. As the incidence of hyperhidrosis was lower with advanced age, individuals aged 18–65 years were included in the analysis [20]. In order to match hyperhidrosis patients with control subjects, propensity scores were calculated using a logistic regression model based on the following variables: Age, sex, and comorbidities (diabetes mellitus [E10–E14], hypertension [I10–I15], atrial fibrillation [I48], dyslipidemia [E78], congestive heart failure [I50], mood disorder [F30–F39], anxiety disorder [F40–F41], renal disease [N17–N19], and malignant neoplasm [C00–D48]). Further analysis was performed by dividing hyperhidrosis patients into those who underwent ETS and those who did not. Subjects were followed up until the first cardiovascular event or 31 December 2017. A flowchart of the study design is presented in Figure 1.

### 2.3. Statistical Analyses

Descriptive statistics were calculated to analyze the control subjects and hyperhidrosis patients who underwent ETS and those who did not. The characteristics of study subjects were compared using t-test and one-way analysis of variance for continuous variables and chi-squared test for categorical variables. Mean and proportion differences between groups were determined using post-hoc analysis of variance and chi-squared test with Bonferroni corrections. Kaplan–Meier curves were constructed, and comparisons were performed using the log-rank test. We calculated the hazard ratio (HR) and 95% confidence interval (95% CI) with Cox proportional hazards regression analysis. Statistical significance was set at p-value < 0.05 (overall), and all statistical analyses were performed using SAS statistical software (version 9.4.2; SAS Institute Inc., Cary, NC, USA).

## 3. Results

With a median follow-up time of 7.7 years, 33,746 patients in total were diagnosed with hyperhidrosis in 2010, whereas 45,125,621 individuals were not. Following the exclusion of hyperhidrosis patients with cardiovascular events (stroke and ischemic heart diseases) prior to 2010, 19,475 patients were deemed to have hyperhidrosis, whereas 45,432,866 individuals were not (controls). Among these individuals, those aged 18–65 years were included in the analysis, resulting in a total of 18,613 hyperhidrosis patients and 29,267,138 controls. After propensity score matching by age, sex, and comorbidities, the study cohort comprised 18,613 hyperhidrosis patients and 18,613 controls (Figure 1).

Table 1 presents the baseline characteristics of the study subjects. No significant differences in mean age, sex, and comorbidities were observed between the control and the hyperhidrosis groups. Among 18,613 hyperhidrosis patients, 2404 patients underwent ETS, whereas 16,209 patients did not. During the follow-up period, 462 and 571 cardiovascular events were identified in the control and hyperhidrosis groups, respectively (Figure 1).

The cumulative incidence of cardiovascular events including stroke and ischemic heart diseases was significantly higher in the hyperhidrosis group than in the control group (log-rank *p* = 0.002; Figure 2). The results of univariate and multivariate Cox regression analyses for the risk of stroke, ischemic heart disease, and combined cardiovascular events are shown in Table 2. The hyperhidrosis group had a significantly higher risk for combined cardiovascular events than the control group (HR, 1.24; 95% CI, 1.10–1.41) after adjustment for age, sex, and comorbidities.

However, no significant difference in the cumulative incidence of cardiovascular events including stroke and ischemic heart diseases was observed between the control group and hyperhidrosis patients who underwent ETS. Furthermore, the cumulative incidence of combined cardiovascular events was higher in hyperhidrosis patients who did not undergo ETS than in the control group (log-rank *p* < 0.001; Figure 3). Table 3 presents the HR for cardiovascular events according to ETS status determined through Cox regression analysis. Hyperhidrosis patients who did not undergo ETS had a significantly higher risk for combined cardiovascular events than control patients (HR, 1.28; 95% CI, 1.13–1.45) after adjustment for age, sex, comorbidities. However, the risk of combined cardiovascular events did not significantly differ between hyperhidrosis patients who underwent ETS and the control group (HR, 0.89; 95% CI, 0.63–1.26).

## 4. Discussion

Hyperhidrosis is characterized by excessive sweating beyond the thermoregulatory requirements, and several patients suffer from this condition, resulting in a negative impact on their quality of life owing to emotional, physical, or social discomfort. Although the etiology of primary hyperhidrosis remains obscure, an increase in sympathetic activity and autonomic imbalance in hyperhidrosis patients have recently been reported [21]. In addition, sympathectomy for hyperhidrosis patients has been shown to confer cardiovascular benefits [17,22,23]. Nakamura et al. reported that ETS decreased the myocardial oxygen demand via a reduction in heart rate and arterial pressure [16]. Cruz et al. demonstrated that ETS increased cardiac autonomic activity [22]. Recently, Cheng et al. have shown that ETS reduced the risk of cardiovascular events in hyperhidrosis patients [17], which is consistent with the results of our study. However, there exist few previous studies to allow for a comparison of the incidence and risk of cardiovascular events between individuals with and without hyperhidrosis; thus, it remains unknown whether hyperhidrosis increases the risk of cardiovascular diseases. To the best of our knowledge, our study is the first to compare cardiovascular outcomes between individuals with and without hyperhidrosis using nationwide population-based cohort data. We determined that hyperhidrosis patients had a significantly higher risk for cardiovascular events than individuals without hyperhidrosis. However, when we divided cardiovascular events into stroke and ischemic heart diseases, the risk for ischemic heart diseases in the hyperhidrosis group did not significantly differ from that in the control group. As the hyperhidrosis group included both hyperhidrosis patients who did and those who did not undergo ETS, it is assumed that the overall effect of hyperhidrosis on ischemic heart diseases might have been weakened. In our study, hyperhidrosis patients who did not undergo ETS had a significantly higher risk for combined cardiovascular events than the control subjects. On the contrary, hyperhidrosis patients who underwent ETS and control subjects showed comparable cardiovascular outcomes.

The exact underlying mechanisms for the relationship between hyperhidrosis and cardiovascular events remain uncertain; nevertheless, chronic sympathetic overactivity plays a key pathophysiological role in both conditions. Thermoregulation through sweat production is controlled by cerebral cortical structures, the preoptic region of the anterior hypothalamus [24], and the sympathetic nervous system [25]. Because the sweat glands are innervated by sympathetic cholinergic nerve fibers [4], the activation of the sympathetic nervous system could stimulate sweat secretion, and the sweat glands could respond to catecholamines in emotionally induced sweating [26,27]. Furthermore, sympathetic overactivity could increase the catecholamine-mediated modulation of heart rate, vasomotor tone [11,28], and platelet aggregability [29] and contribute to the activation of the renin–angiotensin–aldosterone system, subsequently stimulating myocardial contractility, constricting peripheral arteries, and promoting heart failure and atherosclerosis [11,30,31,32].

Noninvasive strategies such as behavioral changes and antiperspirant use are considered first-choice treatments for hyperhidrosis; however, surgery can be considered for severe hyperhidrosis patients who fail to respond to conservative treatments. ETS has been regarded as the intervention of choice for primary hyperhidrosis to reduce sympathetic overactivity owing to its high success rate, low morbidity, relative ease of accessibility, and short recovery time required [33]. Hyperhidrosis patients who undergo ETS occasionally have the potential risk of compensatory sweating, the most common side effect. However, growing evidence has shown that ETS as a treatment for hyperhidrosis has additional cardiovascular protective effects. Recent studies have reported a decrease in heart rate and sympathetic activity and an increase in vagal and cardiac autonomic activities among patients who underwent ETS [22,34]. Jeng et al. reported increased carotid artery flow volume, increased middle cerebral artery flow velocity, and reduced blood pressure in hyperhidrosis patients after ETS [23]. In keeping with these studies, our study showed that hyperhidrosis patients who did not undergo ETS had a higher risk for combined cardiovascular events including stroke and ischemic heart diseases than the control group. However, no significant difference in the risk of combined cardiovascular events including stroke and ischemic heart diseases was observed between the hyperhidrosis patients who underwent ETS and the control group.

Our study has several limitations. First, the HIRA database provides limited information about patients (age, sex, diagnostic date, diagnostic codes, and procedure codes) and does not include the entire medical records. Therefore, we could not exactly estimate the cardiovascular risks and events in hyperhidrosis patients, and cardiovascular event prevalence and incidence rates could have been possibly overestimated. Second, the HIRA database does not contain detailed data on hyperhidrosis characteristics such as the location, severity, duration, and presence of compensatory hyperhidrosis; laboratory and radiologic findings; socioeconomic status (educational level and income); and other clinical considerations that may potentially affect cardiovascular events (e.g., alcohol consumption, smoking status, body mass index, physical activity, family history). Third, we did not match ETS patients versus non-ETS patients by propensity score. However, to minimize this limitation, we adjusted significant covariates, including age, sex, and variable comorbidities that might potentially affect cardiovascular events (diabetes mellitus, hypertension, atrial fibrillation, dyslipidemia, congestive heart failure, mood disorder, anxiety disorder, renal disease, malignant neoplasm) in the Cox regression analysis. Fourth, we did not divide stroke outcomes into ischemic and hemorrhagic stroke. A previous study reported that although hyperhidrosis patients who underwent ETS showed a reduced risk of ischemic stroke, ETS was not associated with a reduction in hemorrhagic stroke [35]. Further studies are warranted to establish the effects of hyperhidrosis according to stroke types (ischemic and hemorrhagic stroke). Lastly, because our study is not a randomized controlled trial, it is possible that residual confounding factors remain, and we could not confirm the causal relationship between hyperhidrosis and cardiovascular diseases. Despite these potential limitations, we performed a large population-based cohort study with a relatively long duration, covering >99% of the South Korean population. In addition, to minimize confounding factors, age, sex, and well-known cardiovascular comorbidities were matched by propensity scores and adjusted for in the Cox regression analysis.

## 5. Conclusions

Hyperhidrosis patients showed a significantly higher risk for cardiovascular events than individuals without hyperhidrosis, and ETS could reduce the risk of cardiovascular events in hyperhidrosis patients. Physicians should pay attention to potential cardiovascular risks when treating hyperhidrosis patients, and ETS needs to be considered for high-risk patients with cardiovascular diseases despite the presence of compensatory hyperhidrosis and potential complications of ETS.

## Figures and Tables

**Figure 1 ijerph-16-03925-f001:**
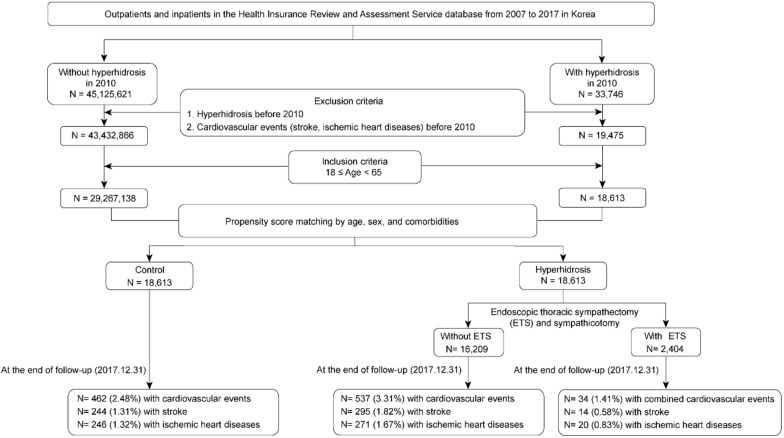
Flowchart of subject selection from the Health Insurance Review and Assessment Service database of Korea.

**Figure 2 ijerph-16-03925-f002:**
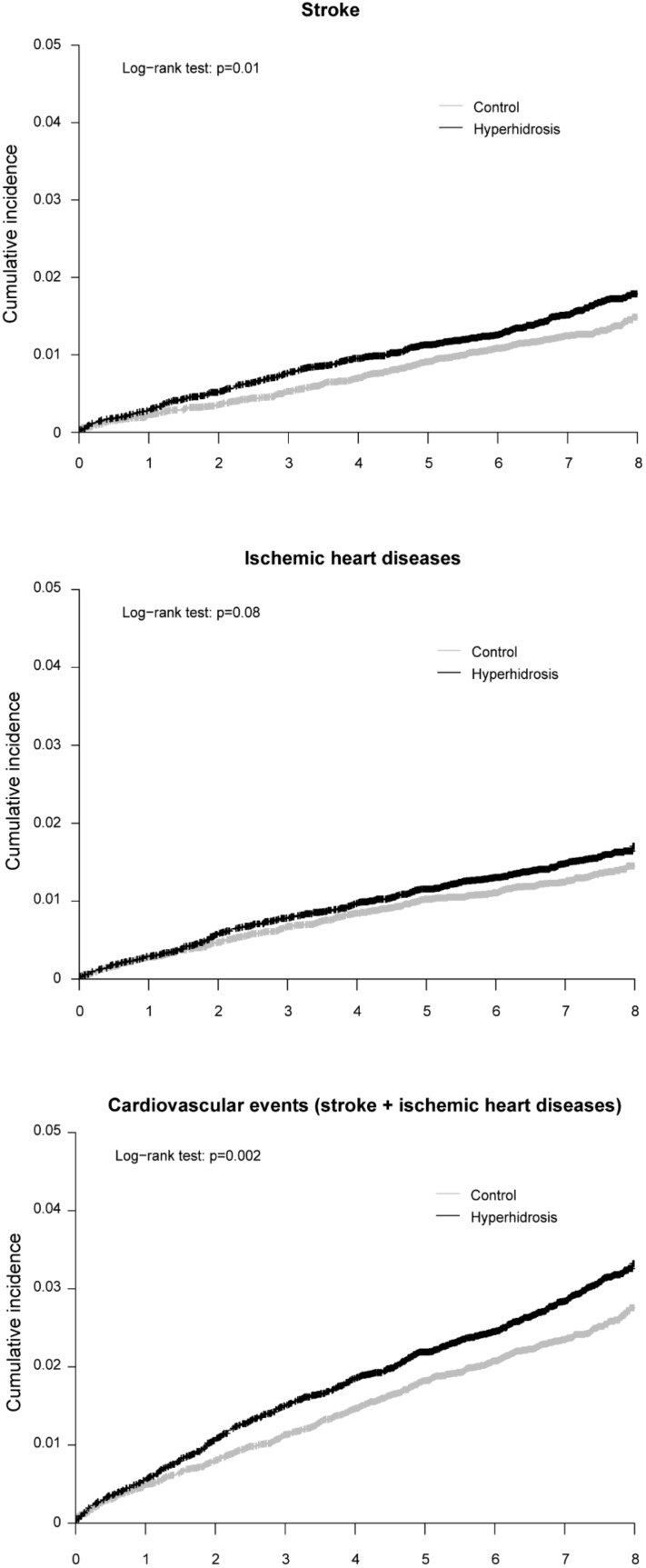
Kaplan–Meier curves with cumulative incidence of cardiovascular events stratified by the presence or absence of hyperhidrosis using log-rank test.

**Figure 3 ijerph-16-03925-f003:**
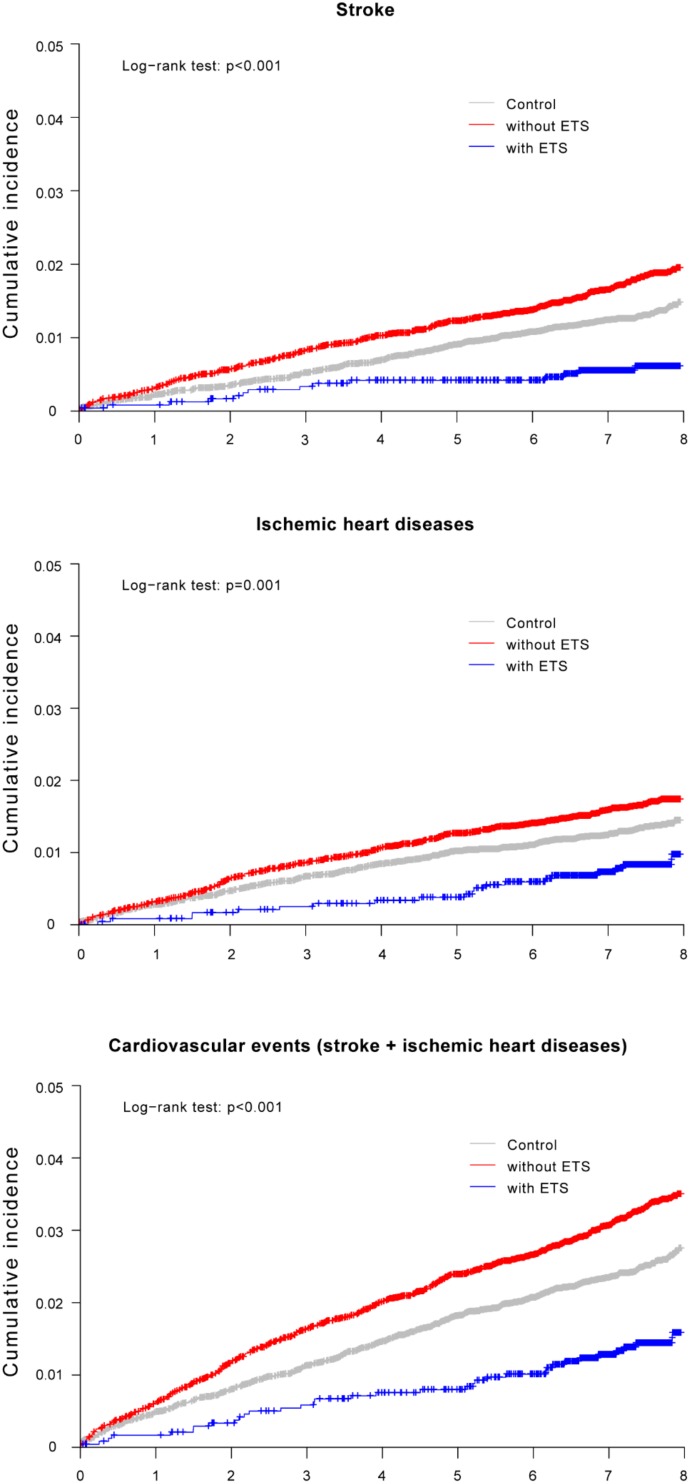
Kaplan–Meier curves with cumulative incidence of cardiovascular events stratified by the presence or absence of hyperhidrosis and Endoscopic thoracic sympathectomy (ETS) status using log-rank test.

**Table 1 ijerph-16-03925-t001:** Baseline characteristics of the study subjects according to the presence or absence of hyperhidrosis and ETS status.

	Control	Hyperhidrosis	*p*-Value	Control	Hyperhidrosis		*p*-Value	Post Hoc
					Without ETS	With ETS		
N	18,613	18,613		18,613	16,209	2404		
Age (years)	32.8 ± 12.2	32.8 ± 12.2	0.961	32.8 ± 12.2	33.4 ± 12.4	28.8 ± 9.9	<0.001	a, b, c
Sex			0.909				0.041	b, c
Female	9123 (49.0%)	9134 (49.1%)		9123 (49.0%)	8012 (49.4%)	1122 (46.7%)		
Male	9490 (51.0%)	9479 (50.9%)		9490 (51.0%)	8197 (50.6%)	1282 (53.3%)		
Comorbidity								
Diabetes mellitus	640 (3.4%)	633 (3.4%)	0.842	640 (3.4%)	609 (3.8%)	24 (1.0%)	<0.001	b, c
Hypertension	1301 (7.0%)	1292 (7.0%)	0.855	1301 (7.0%)	1204 (7.4%)	88 (3.7%)	<0.001	b, c
Atrial fibrillation	47 (0.3%)	37 (0.2%)	0.275	47 (0.3%)	34 (0.2%)	3 (0.1%)	0.394	—
Dyslipidemia	1185 (6.4%)	1202 (6.5%)	0.719	1185 (6.4%)	1096 (6.8%)	106 (4.4%)	<0.001	b, c
Congestive heart failure	22 (0.1%)	22 (0.1%)	>0.999	22 (0.1%)	21 (0.1%)	1 (0.1%)	0.504	—
Mood disorder	862 (4.6%)	853 (4.6%)	0.824	862 (4.6%)	785 (4.8%)	68 (2.8%)	<0.001	b, c
Anxiety disorder	809 (4.4%)	809 (4.4%)	>0.999	809 (4.4%)	733 (4.5%)	76 (3.2%)	0.009	b, c
Renal disease	46 (0.2%)	31 (0.2%)	0.087	46 (0.2%)	30 (0.2%)	1 (0.1%)	0.081	—
Malignant neoplasm	243 (1.3%)	225 (1.2%)	0.402	243 (1.3%)	210 (1.3%)	15 (0.6%)	0.016	b, c

Data are presented as mean ± standard deviation or N (percentage); p-values were calculated using t-test or one-way analysis of variance for continuous variables and chi-squared test for categorical variables. Post-hoc analysis of variance for the mean difference between groups: a, control vs. hyperhidrosis patients who did not undergo ETS; b, control vs. hyperhidrosis patients who underwent ETS; c, hyperhidrosis patients who did not undergo ETS vs. hyperhidrosis patients who underwent ETS. ETS: Endoscopic thoracic sympathectomy.

**Table 2 ijerph-16-03925-t002:** Hazard ratio for cardiovascular events according to the presence or absence of hyperhidrosis determined through Cox regression analysis.

	Control	Hyperhidrosis
Stroke		
Crude HR (95% CI)	1 (reference)	1.24 (1.05–1.47)
Adjusted HR ^1^ (95% CI)	1 (reference)	1.28 (1.08–1.51)
Ischemic heart diseases		
Crude HR (95% CI)	1 (reference)	1.16 (0.98–1.38)
Adjusted HR ^1^ (95% CI)	1 (reference)	1.17 (0.99–1.39)
Combined cardiovascular events		
Crude HR (95% CI)	1 (reference)	1.22 (1.08–1.37)
Adjusted HR ^1^ (95% CI)	1 (reference)	1.24 (1.10–1.41)

^1^ Adjusted for age, sex, and comorbidities (diabetes mellitus, hypertension, atrial fibrillation, dyslipidemia, congestive heart failure, mood disorder, anxiety disorder, renal disease, malignant neoplasm). HR: Hazard ratio; CI: Confidence interval.

**Table 3 ijerph-16-03925-t003:** Hazard ratio for cardiovascular events according to ETS status determined through Cox regression analysis.

	Control	Hyperhidrosis	
		Without ETS	With ETS
Stroke			
Crude HR (95% CI)	1 (reference)	1.36 (1.15–1.61)	0.44 (0.26–0.75)
Adjusted HR ^1^ (95% CI)	1 (reference)	1.32 (1.12–1.57)	0.72 (0.42–1.24)
Ischemic heart diseases			
Crude HR (95% CI)	1 (reference)	1.24 (1.05–1.48)	0.62 (0.39–0.98)
Adjusted HR ^1^ (95% CI)	1 (reference)	1.19 (1.01–1.41)	0.97 (0.61–1.53)
Combined cardiovascular events			
Crude HR (95% CI)	1 (reference)	1.31 (1.16–1.49)	0.56 (0.40–0.79)
Adjusted HR ^1^ (95% CI)	1 (reference)	1.28 (1.13–1.45)	0.89 (0.63–1.26)

^1^ Adjusted for age, sex, and comorbidities (diabetes mellitus, hypertension, atrial fibrillation, dyslipidemia, congestive heart failure, mood disorder, anxiety disorder, renal disease, malignant neoplasm).

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
