# Peer review of "Hyperhidrosis, Endoscopic Thoracic Sympathectomy, and Cardiovascular Outcomes: A Cohort Study Based on the Korean Health Insurance Review and Assessment Service Database"

_ijerph, 2019, doi:10.3390/ijerph16203925_

Round 1
Reviewer 1 Report
To the Authors:
This is a very good manuscript with a new approach to this entity, hyperhidrosis.
My only concern is in Table 3. Since the Adjusted HR for Stroke in Hyperhidrosis patients with ETS is 0.72 and for Ischemic heart diseases is 0.97, is it correct that the Combined cardiovascular events is 0.98? Could you please check this number?
Congratulations.
Author Response
We appreciate the reviewer’s insightful comments regarding our manuscript. We had mistakenly written 0.89 as 0.98; 0.98 is a typographical error, whereas 0.89 is correct. We have accordingly corrected 0.98 to 0.89 in Table 3, as follows:
Table 3. Hazard ratio for cardiovascular events according to ETS status determined through Cox regression analysis.
|
Control |
Hyperhidrosis |
||
|
without ETS |
with ETS |
||
|
Stroke |
|||
|
Crude HR (95% CI) |
1 (reference) |
1.36 (1.15–1.61) |
0.44 (0.26–0.75) |
|
Adjusted HR 1 (95% CI) |
1 (reference) |
1.32 (1.12–1.57) |
0.72 (0.42–1.24) |
|
Ischemic heart diseases |
|||
|
Crude HR (95% CI) |
1 (reference) |
1.24 (1.05–1.48) |
0.62 (0.39–0.98) |
|
Adjusted HR 1 (95% CI) |
1 (reference) |
1.19 (1.01–1.41) |
0.97 (0.61–1.53) |
|
Combined cardiovascular events |
|||
|
Crude HR (95% CI) |
1 (reference) |
1.31 (1.16–1.49) |
0.56 (0.40–0.79) |
|
Adjusted HR 1 (95% CI) |
1 (reference) |
1.28 (1.13–1.45) |
0.89 (0.63–1.26) |
1 Adjusted for age, sex, and comorbidities (diabetes mellitus, hypertension, atrial fibrillation, dyslipidemia, congestive heart failure, mood disorder, anxiety disorder, renal disease, malignant neoplasm).
ETS: endoscopic thoracic sympathectomy; HR: hazard ratio; CI: confidence interval

Reviewer 2 Report
We would like to congratulate with the authors for the study.
In addition to those already underlined by the Authors, there are other limitations:
Is a retrospective study. Focusing on the characteristics of this study is very original the idea to correlate and indagate the indreased risk of cardiovascular diseases in patients affected by hyperhidrosis. On the opposite it is equally true that the comparison between patient with hyperhidrosis treated with surgery and not treated is already investigated in a recent Taiwanese study during the 2018 (Cheng CG et all. Associated with Ischemic Stroke Risk Reduction after Endoscopic Thoracic Sympathectomy for Palmar Sweating J Stroke Cerebrovasc Dis. 2018 Aug;27(8):2235-2242. ) The control group has a sampling size of 18613 patients; on the contrary, the Hyperidrosis group is divided in two subgroups with a significant difference in the sampling size (without ETS 16309 vs with ets 2404). For these reasons is manifestly clear that there is a strong inhomogeneity due to the characteristic of the two groups and is not recommended to place to comparison when the sampling size is clearly different and one of the two groups is ulteriorly divided in two subcategories.
For these reasons the study can’t be accepted.
Author Response
Thank you for these insightful comments. We admit that the hyperhidrosis group was divided in two subgroups, with a significant difference in the sampling size. Nonetheless, we aimed to compare the incidence and risk of cardiovascular events between individuals with and without hyperhidrosis, and we found that hyperhidrosis patients were at a significantly higher risk for cardiovascular events than individuals without hyperhidrosis. This finding was not investigated in previous studies by Cheng et al. We did not match ETS versus non-ETS by propensity score but instead focused on matching individuals with and without hyperhidrosis by propensity score. Although we did not match ETS versus non-ETS by propensity score, we adjusted significant covariates, including age, sex, and variable comorbidities that might potentially affect cardiovascular events (diabetes mellitus, hypertension, atrial fibrillation, dyslipidemia, congestive heart failure, mood disorder, anxiety disorder, renal disease, malignant neoplasm), in order to minimize this limitation. In view of your comments, we have added the following paragraph in the Discussion section to serve as one of our study’s limitations:
|
Discussion section, page 10, lines 252–256 Third, we did not match ETS versus non-ETS by propensity score. However, to minimize this limitation, we adjusted significant covariates, including age, sex, and variable comorbidities that might potentially affect cardiovascular events (diabetes mellitus, hypertension, atrial fibrillation, dyslipidemia, congestive heart failure, mood disorder, anxiety disorder, renal disease, malignant neoplasm) in the Cox regression analysis. |

Reviewer 3 Report
Dear Authors
I found your paper very interesting since addressing a not well known topic about hyperhidrosis and its management. Despite several Authors have already reported their results in hyperhidrosis management, just few investigated influence on cardiovascular events risk. Therefore, these data will be absolutely useful for further study or even for clinical practice. As you mentioned main limitation is about quality of information provided by your database since based on codes. That is why I suggest you to compare results with similar papers in literature in order to provide a broader overview of the topic. Therefore, after minor revision, the paper should be accepted for publication.
Author Response
Thank you for your valuable comments. However, there exist few previous studies to allow for a comparison of the incidence and risk of cardiovascular events between individuals with and without hyperhidrosis. To the best of our knowledge, our study is the first to compare cardiovascular outcomes between individuals with and without hyperhidrosis. We have revised the Discussion section as follows to compare our results with those of similar studies in the literature in order to provide a broader overview of the topic:
|
Discussion section, page 9, lines 198–205 Nakamura et al. reported that ETS decreased the myocardial oxygen demand via a reduction in heart rate and arterial pressure [16]. Cruz et al. demonstrated that ETS increased cardiac autonomic activity [22]. Recently, Cheng et al. have shown that ETS reduced the risk of cardiovascular events in hyperhidrosis patients [17], which is consistent with the result of our study. However, there exist few previous studies to allow for a comparison of the incidence and risk of cardiovascular events between individuals with and without hyperhidrosis; thus, it remains unknown whether hyperhidrosis increases the risk of cardiovascular diseases. |

Round 2
Reviewer 2 Report
Now the paper is complete. It is ready for publication. I want to remember that Hyperhidrosis is a hot theme always.